# DARE–Agent: Domain–Aware, Resource–Efficient, Evidence–grounded Agentic RAG

## Abstract

LLM agents for deep research have advanced open–domain reasoning, yet deployments in specialized domains still fail along three critical axes: unverifiable answers, uncontrolled cost, and domain–agnostic retrieval that undermines authority/recency. Prevailing evaluations focus narrowly on answer accuracy, overlooking process–level metrics such as citation precision (CP), minimal sufficient evidence (MSE), and the accuracy–cost trade–off, while many training setups rely on complex, hard–to–reproduce online RL. We reframe research–agent quality as a multi–objective problem spanning accuracy, verifiability, and resource efficiency, and introduce **DARE–Agent**, a domain–aware, resource–efficient, evidence–grounded agentic RAG framework. DARE–Agent integrates a learnable domain–aware gating mechanism into a short, auditable trajectory: the agent proposes domain–conditioned controls over retrieval and evidence, and an executor clips them to safe ranges. Training combines SFT with Direct Preference Optimization over multiple sampled trajectories using a composite preference that balances accuracy, verifiability, cost, and redundancy; retrieved tokens are loss–masked for stability. In a reproducible fixed–corpus setting plus small live–web subsets, DARE–Agent delivers competitive accuracy while consistently improving citation precision, reducing MSE, and yielding stronger accuracy–cost Pareto fronts under matched budgets; it also raises authority/recency hit rates.

## 1 Introduction

Large language models (LLMs) augmented with retrieval have become a standard recipe for knowledge-intensive reasoning. Early retrieval-augmented systems ground generation on external corpora to reduce hallucination and improve factuality (Lewis et al., 2020; Izacard & Grave, 2021; Shuster et al., 2021), while agentic variants interleave tool use and reasoning to plan, browse, and cite sources (Yao et al., 2023; Nakano et al., 2021; Tan et al., 2025). Despite this progress, deployments in specialized domains (biomedicine, law, finance) still fail along three practical axes: unverifiable answers with weak or redundant evidence, uncontrolled cost from excessive tool calls and token usage, and domain-agnostic retrieval that overlooks authority and recency signals.

These limitations are compounded by evaluations that emphasize answer accuracy alone, with limited attention to process-level metrics such as citation correctness, evidence sufficiency, and the accuracy–cost trade-off (Liang et al., 2022). Meanwhile, many tool-use training paradigms rely on complex, hard-to-reproduce online reinforcement learning. Our goal is to advance agentic retrieval-augmented generation toward verifiable, resource-efficient behavior that adapts to domain constraints, without the engineering overhead of online RL.

We reframe research-agent quality as a multi-objective problem spanning accuracy, verifiability, and resource efficiency, aligning with calls for holistic, multi-metric evaluation (Liang et al., 2022). Beyond exactness of answers, we emphasize process desiderata: citation precision and minimal sufficient evidence (MSE), measured alongside compute and latency budgets. While recent work adapts retrieval frequency or integrates self-reflection to improve factuality (Asai et al., 2023; Shinn et al., 2023), rigorous measurement and optimization of these process metrics remain underreported.

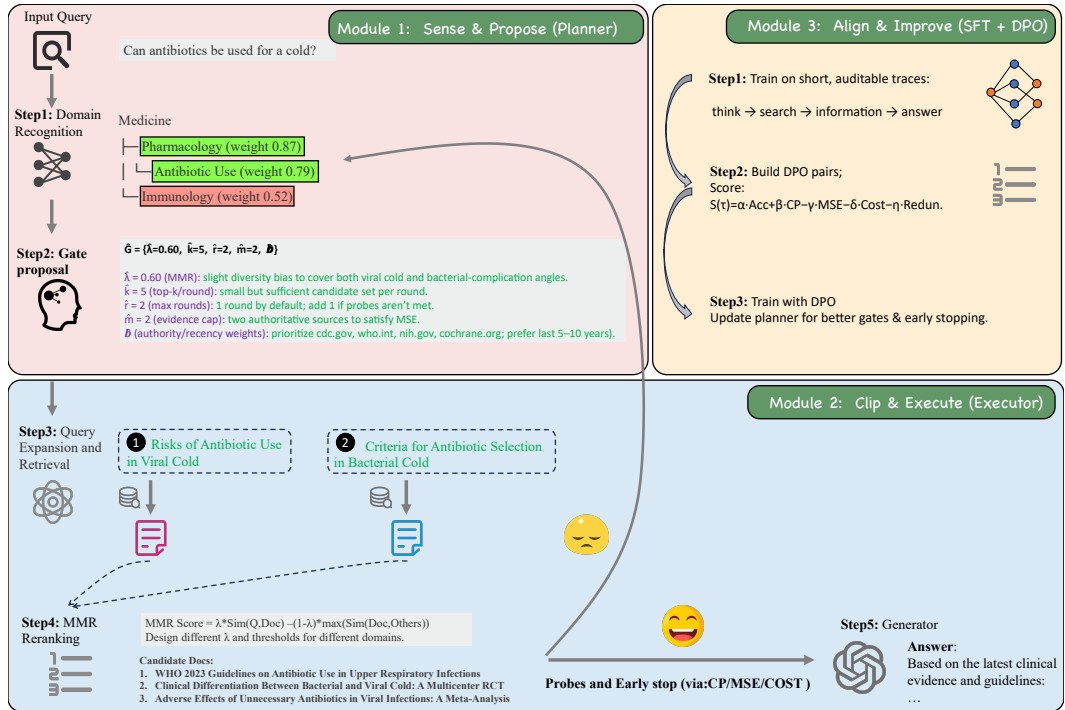

Figure 1: DARE-Agent framework (SAP–CAE–AAI). Module 1 (SAP: Sense & Propose) detects the domain (e.g., Medicine) and proposes gates $\hat{g} = \{\hat{\lambda} = 0.60, \hat{k} = 5, \hat{r} = 2, \hat{m} = 2, \hat{b}\}$ to control retrieval breadth, rounds, citation budget, and authority/recency weights. Module 2 (CAE: Clip & Execute) clips to safe bounds, expands sub-queries, performs hybrid retrieval with MMR ($\lambda = 0.60$), and answers with citations once citation precision and minimal sufficient evidence (MSE) thresholds are met; otherwise it loops back to SAP. Module 3 (AAI: Align & Improve) improves the planner/generator via SFT warm-start and DPO on logged trajectories with composite preference $S(\tau) = \alpha \cdot \text{Acc} + \beta \cdot \text{CP} - \gamma \cdot \text{MSE} - \delta \cdot \text{Cost} - \eta \cdot \text{Redun}$.

**Scope and Positioning.** DARE-Agent targets a *reproducible, RL-free* path to agent alignment under matched budgets, with *short, auditable trajectories* and *explicit domain-aware gates*. Our objective is not to chase peak single-metric accuracy, but to deliver stronger *Pareto fronts* over (accuracy, verifiability, cost). During inference we use *lightweight CP/MSE proxies* to guide early stopping (their computation is accounted in Cost), while a *frozen LLM-as-a-judge* is used *only for offline evaluation* and does not contribute to inference-time costs (details in Sec. 3.4, App. D).

To this end, we introduce **DARE-Agent**, a Domain-Aware, Resource-Efficient, evidence-grounded agentic RAG framework (Figure 1). DARE-Agent executes a short, auditable trajectory–*think → search → information* (repeat under a budget) *→ answer*–and operationalizes domain awareness via a learnable domain-aware gating mechanism. In the *think* step, the model proposes domain-conditioned controls over retrieval and evidence, including the MMR trade-off $\lambda$, top-$k$, the number of search rounds, a cap on evidence items, and authority/recency allowlists; a lightweight executor then clips these proposed gates to safe ranges (*propose-and-clip*), logs the proposal/execution pair, and updates cost counters.

Training eschews online RL in favor of a reproducible post-training pipeline: supervised fine-tuning (SFT) for cold-start trajectories, followed by Direct Preference Optimization (DPO) (Rafailov et al., 2023). For each question, we sample multiple trajectories that vary the domain-aware gates and stopping decisions, compute a composite preference score that balances accuracy, verifiability (citation precision and MSE), cost (tokens and latency), and redundancy, and construct preference pairs for DPO. Retrieved content is loss-masked to prevent label leakage and to stabilize optimization. The resulting single model learns when to retrieve, how much to retrieve, which sources to trust, and when to stop without an explicit reward model or a policy-gradient loop.

We evaluate DARE-Agent in two regimes. (1) A reproducible fixed-corpus setting spanning open-domain QA and specialized domains (Natural Questions, HotpotQA, 2WikiMultiHopQA, Pub-MedQA) using shared indexes and a common retriever (Kwiatkowski et al., 2019; Yang et al., 2018; Ho & et al., 2020; Jin et al., 2019). (2) Agentic browsing environments that stress multi-step search and tool use (GAIA, WebArena) (Mialon et al., 2023; Zhou et al., 2023). Across datasets and under matched budgets, DARE-Agent delivers competitive answer accuracy while consistently improving citation precision, lowering MSE, and yielding stronger accuracy–cost Pareto fronts.

**Contributions.**

- A domain-aware agentic RAG framework with a *propose-and-clip* gating mechanism that turns domain identification into explicit, learnable controls over retrieval, evidence selection, and budgets within a short, auditable trajectory.
- An RL-free post-training recipe (SFT+DPO on composite preferences) that aligns agents to multi-objective quality (accuracy, verifiability, and cost), with loss masking over retrieved content for stability.
- A reproducible evaluation protocol with probes for citation precision, MSE, and cost, applied to fixed-corpus QA and agentic browsing benchmarks, demonstrating improved accuracy–cost Pareto fronts under matched budgets.

## 2 RELATED WORK

We position DARE-Agent within four threads: retrieval-augmented generation, agentic tool use and web browsing, evaluation and verifiability, and preference-based alignment.

**Retrieval-augmented generation (RAG).** Early work integrates retrieval to ground generation and reduce hallucination, including pre-training with latent retrieval (REALM) and encoder-decoder architectures that condition on retrieved passages (RAG, FiD). Subsequent approaches add critique or self-reflection to improve factuality, e.g., Self-RAG and Reflexion (Asai et al., 2023; Shinn et al., 2023). Our framework shares the grounding premise but differs in (i) converting domain awareness into explicit, learnable gates for retrieval intensity and evidence breadth, (ii) incorporating authority and recency signals during ranking, and (iii) optimizing for process-level quality (citation precision and minimal sufficient evidence) under explicit cost budgets.

**Agentic reasoning and web-enabled QA.** Interleaving reasoning with tool use has been explored by ReAct and WebGPT (Yao et al., 2023; Nakano et al., 2021). RL-based browsing agents (e.g., Search-R1, DeepResearcher) learn interleaved search-and-reasoning with outcome rewards in static corpora or the live web (Jin et al., 2025; Zheng et al., 2025). DARE-Agent adopts short, auditable trajectories with a propose-and-clip executor that enforces domain/budget constraints, trading some peak accuracy for verifiability/efficiency and easier reproducibility. We also discuss MedResearcher-R1 (domain tools) and WebWatcher (multimodal) as complementary directions (Yu et al., 2025; Geng et al., 2025).

**Preference-based alignment and RL-free optimization.** Beyond RLHF, DPO/SimPO provide RL-free alternatives for preference alignment (Rafailov et al., 2023; Meng et al., 2024). We choose RL-free post-training for *reproducibility, cost/stability,* and *explicit multi-objective control* (CP/M-SE/Cost), while RL-based agents remain a strong complementary line (Jin et al., 2025; Zheng et al., 2025).

**Fairness of comparisons.** Online RL browsing agents typically operate under different interaction budgets and environments; aligning every knob is non-trivial. We therefore standardize a *matched-budget, fixed-corpus/browsing* protocol shared by all compared methods in this paper, and place non-budget-matched RL results (if any) in the appendix strictly as contextual references.

## 3 METHODS

We cast agentic retrieval-augmented generation (RAG) as a short, auditable decision process that jointly controls when to retrieve, how much to retrieve, which sources to trust, and when to stop.

DARE-Agent operationalizes this with three components: (i) domain-aware *propose-and-clip* gating (Sec. 3.2); (ii) authority-/recency-aware retrieval with diversity control (Sec. 3.3); and (iii) an RL-free post-training pipeline (SFT → DPO) with evidence-span loss masking (Sec. 3.5). Throughout, we relate our design to the ReAct act-and-reason paradigm (Yao et al., 2023) and grounding/evaluation practices, including LLM-as-a-judge (Zheng et al., 2023).

## 3.1 PROBLEM SETUP AND NOTATION

Given a query $q$, the agent interacts with a retriever and produces an answer $y$ with cited evidence $E = \{e_i\}_{i=1}^m$. Let $d \in \mathcal{D}$ denote a domain inferred in-context during planning (e.g., biomedicine, law). At step $t$, the agent state $s_t$ composes the dialogue prefix, the current plan, accumulated evidence $E_{1:t}$, and running cost counters (tool calls, retrieved/generation tokens, latency).

The planner $\pi_\theta$ proposes a vector of *gates*

$$\hat{g}_t = \big(\hat{\lambda}_t, \ \hat{k}_t, \ \hat{r}_t, \ \widehat{m}_t, \ \widehat{\mathbf{b}}_t\big), \tag{1}$$

where $\lambda \in [0, 1]$ is the MMR trade-off, $k$ is the per-round top-$k$, $r$ is the remaining search rounds, $m$ is the cap on evidence items, and $\mathbf{b}$ encodes domain-conditioned priors over sources (an *allowlist* with nonnegative weights). The executor clips $\hat{g}_t$ to safe ranges and enforces budgets.

The agent optimizes three objectives: answer accuracy, verifiability (citation precision/sufficiency and minimal sufficient evidence), and resource efficiency (cost in tool calls/tokens/latency). These are surfaced both at inference (through $g_t$ and stopping) and during learning (through composite preferences).

## 3.2 DOMAIN-AWARE GATING VIA PROPOSE-AND-CLIP

DARE-Agent turns domain awareness into explicit, learnable control signals over retrieval and evidence.

**Propose.** A planning model $\pi_\theta$ generates a structured plan containing $(d_t, \hat{g}_t)$ conditioned on $s_t$:

$$(d_t, \hat{g}_t) \sim \pi_\theta(\cdot \mid s_t). \tag{2}$$

**Clip and log.** A lightweight executor enforces domain- and budget-specific constraints via element-wise clipping,

$$g_t = \mathrm{clip}\big(\hat{g}_t; \ g^{\min}(d_t), \ g^{\max}(d_t)\big), \tag{3}$$

logs $(\hat{g}_t \to g_t)$ for auditability, updates cost counters, and halts if any budget is exceeded. Intuitively, this *propose-and-clip* gate allows the model to *suggest* retrieval intensity and evidence breadth, while a deterministic layer guarantees safety and reproducibility.

**Stopping.** The agent answers if (i) rounds are exhausted or a budget is met, or (ii) the estimated support is strong enough: $\widehat{\mathrm{CP}} \geq \tau_c$ and $\widehat{\mathrm{MSE}} \leq \tau_m$ (Sec. 3.4). Otherwise it continues with another search round.

## 3.3 AUTHORITY/RECENCY-AWARE RETRIEVAL WITH DIVERSITY

At each round, a unified retriever returns a candidate pool $\mathcal{C}$ with base similarity $\mathrm{sim}(q, e)$ (dense or hybrid). We refine ranking with domain-aware *authority* and *recency* features to promote credible and timely sources.

**Authority score $A(e; d, \mathbf{b})$.** We define

$$A(e; d, \mathbf{b}) = w_{\mathrm{allow}} \cdot \mathbb{I}\{\mathrm{host}(e) \in \mathrm{Allow}(d, \mathbf{b})\} + w_{\mathrm{host}} \cdot a_{\mathrm{host}}(e; d) + w_{\mathrm{type}} \cdot a_{\mathrm{type}}(e), \tag{4}$$

where (i) $\mathrm{Allow}(d, \mathbf{b})$ is a domain-conditioned allowlist with weights in $\mathbf{b}$; (ii) $a_{\mathrm{host}}(e; d) \in [0, 1]$ is a host-level prior; and (iii) $a_{\mathrm{type}}(e) \in [0, 1]$ reflects document type. Features are min–max normalized per domain. *Initialization.* Allowlist entries and host/type priors are initialized from publicly available domain directories and curated guideline registries (details in App. D); they are *not* hard constraints and can be de-emphasized by $\mathbf{b}$.

**Recency score $R(e)$.** Let $t(e)$ be the publication timestamp (or crawl time). We define $R(e) = \exp(-\mathrm{age}(e)/\tau_d)$ with $\mathrm{age}(e) = t_0 - t(e)$ and domain-specific half-life $\tau_d$ (App. D). When timestamps are missing, we impute from URL patterns and metadata.

**Score fusion and diversity.** Authority and recency adjust the base score:

$$\mathrm{sim}^\star(q, e) = \mathrm{sim}(q, e) + \alpha_a A(e; d_t, \mathbf{b}_t) + \alpha_r R(e), \tag{5}$$

and select via MMR (Carbonell & Goldstein, 1998):

$$e^\star = \arg \max_{e \in \mathcal{C} \setminus S} \Big[ \lambda_t \, \mathrm{sim}^\star(q, e) - (1 - \lambda_t) \max_{e' \in S} \mathrm{sim}(e, e') \Big], \quad S \leftarrow S \cup \{e^\star\}, \tag{6}$$

until $|S| = k_t$ or $m_t$ is reached. Selected snippets are deduplicated, normalized, and appended to $E$ with provenance.

## 3.4 PROCESS PROBES FOR VERIFIABILITY AND COST

We instrument three probes to guide stopping and to supervise preferences.

**Citation precision (CP).** CP is the fraction of cited items deemed supportive by a judgment function $J(\cdot)$. We implement $J$ with either (i) string/entailment heuristics or (ii) an LLM-as-a-judge prompted to label each $(y, e)$ as `support`/`refute`/`irrelevant` (Zheng et al., 2023).

**Minimal sufficient evidence (MSE).** $\mathrm{MSE}(y, E) = \min\{|S| : S \subseteq E, J(S) = \text{support}\}$, approximated via backward elimination with leave-one-out checks under $J$.

**Inference-time proxies vs. offline judging.** At inference, we use lightweight proxies $\widehat{\mathrm{CP}}, \widehat{\mathrm{MSE}}$ (heuristic string/entailment checks) to trigger early stopping; their computation is included in the cost accounting. For reporting, we apply a frozen LLM-as-a-judge offline; this does not affect inference-time behavior or costs (App. D).

**Cost.** We track

$$\mathrm{Cost}(\tau) = \beta_s \, \#\text{search} + \beta_r \, \mathrm{tok}_{\mathrm{retr}} + \beta_g \, \mathrm{tok}_{\mathrm{gen}} + \beta_\ell \, \text{latency}. \tag{7}$$

We normalize costs by the mean cost of the *RAG* baseline on the same split to obtain a dimensionless $\widetilde{\mathrm{Cost}}$ with anchor 1.0 (weights and sensitivity in App. D).

**Redundancy (Redun).** $\mathrm{Redun}(E) = \frac{1}{|E|} \sum_{i=1}^{|E|} \max_{j < i} \cos(\phi(e_i), \phi(e_j))$ using sentence embeddings (BGE-M3).

## 3.5 LEARNING FROM COMPOSITE PREFERENCES: SFT → DPO

**SFT for cold start.** We supervise $\pi_\theta$ on short trajectories that instantiate the think → search → information → answer pattern, including structured gates and explicit citations.

**Composite preference scoring.** For a query $q$, we sample $m$ trajectories $\{\tau_j\}_{j=1}^m$ by varying gates and stopping decisions. We compute

$$S(\tau) = \alpha \, \mathrm{Acc}(\tau) + \beta \, \mathrm{CP}(\tau) - \gamma \, \widehat{\mathrm{MSE}}(\tau) - \delta \, \mathrm{Cost}(\tau) - \eta \, \mathrm{Redun}(\tau), \tag{8}$$

with preference margin $\kappa$ (weights in App. D).

**DPO objective with masking.**

$$\mathcal{L}_{\mathrm{DPO}} = -\mathbb{E}\big[ \log \sigma \big( \beta_{\mathrm{dpo}}(\Delta_\theta - \Delta_{\mathrm{ref}}) \big) \big], \quad \log \pi_\theta(\tau | x) = \sum_t M_t \log \pi_\theta(y_t | y_{<t}, x), \tag{9}$$

where $M_t = 0$ on tokens copied from retrieved spans (evidence-span loss masking). For redundancy, we use BGE-M3; for masking detection we adopt a conservative SBERT 5-gram semantic filter (App. F).

---

**Algorithm 1** DARE-Agent Inference (propose-and-clip)

---

**Require:** query $q$, budgets $(B_{\text{search}}, B_{\text{tok}}, B_\ell)$
1: Initialize $s_1$ with $q$; $E \leftarrow \varnothing$; costs $\leftarrow 0$
2: **for** $t = 1, 2, \ldots$ **do**
3:     $(d_t, \hat{g}_t) \sim \pi_\theta(\cdot \mid s_t)$                                      *propose*
4:     $g_t \leftarrow \text{clip}(\hat{g}_t; g^{\min}(d_t), g^{\max}(d_t))$; update logs/costs          *clip*
5:     **if** rounds exhausted, a budget hit, or $\widehat{\text{CP}} \geq \tau_c$ and $\widehat{\text{MSE}} \leq \tau_m$ **then**
6:         Generate final answer $y$ with citations from $E$; **return** $(y, E)$
7:     **end if**
8:     Retrieve pool $\mathcal{C}$; re-rank with Eq. 4–7; select $S$ via MMR
9:     Normalize and append $S$ to $E$; update $s_{t+1}$ with evidence and costs
10: **end for**

---

## 3.6 INFERENCE AND BUDGETS

Algorithm 1 summarizes inference. Budgets are enforced by the executor; logs include proposed and clipped gates, evidence UUIDs, and probe outputs.

## 3.7 IMPLEMENTATION NOTES AND RELATIONS

We use a unified retriever (dense or hybrid) with cosine similarity; authority and recency features are normalized to $[0, 1]$ and conditioned on domain via **b**. Default gate ranges: $k \in [1, 8], \lambda \in [0.3, 0.8], r \leq 3, m \leq 3$. Sentence-level deduplication precedes MMR. For similarity/Redun we use BGE-M3; for conservative masking in Eq. 7 we use SBERT windowed checks. Optional critique modules (Self-RAG, Reflexion) can be plugged as additional probes without changing the core mechanism.

## 4 EXPERIMENTAL SETUPS

We evaluate DARE-Agent under two complementary regimes: (1) fixed-corpus QA with controlled retrieval, and (2) agentic browsing environments with live tool use. All methods share the same retriever, corpora, budgets, and prompting templates unless stated otherwise.

## 4.1 TASKS AND DATASETS

**Fixed-corpus QA.** We use Natural Questions (NQ), HotpotQA, and 2WikiMultiHopQA (2Wiki) and PubMedQA (Kwiatkowski et al., 2019; Yang et al., 2018; Ho & et al., 2020; Jin et al., 2019). We evaluate with EM/F1 and report process metrics (Sec. 4.7).

**Agentic browsing.** We adopt GAIA and WebArena (Mialon et al., 2023; Zhou et al., 2023). Exact-match is insufficient; we complement task scores with judge-based evaluations and process metrics.

## 4.2 CORPORA, INDEXING, RETRIEVER

We build per-dataset FAISS indexes (checksums in Appendix). Retriever is hybrid: BM25 + dense (BGE-M3). We z-score normalize BM25 per query and linearly fuse with dense cosine:

$$s_{\text{fuse}}(q, e) = w_{\text{bm25}} \tilde{s}_{\text{bm25}}(q, e) + w_{\text{dense}} \cos(h(q), h(e)),$$

with $w_{\text{bm25}} \in [0.3, 0.7]$ grid-searched on dev. Top-$K_0$ from each branch are merged/deduped, re-ranked by authority/recency to obtain $\text{sim}^\star$, then selected by MMR.

## 4.3 MODELS AND INFERENCE

**Planner/Generator:** Qwen2.5-32B-Instruct (ReAct-style schema with `<think>`/`<search>`/`<information>`/`<answer>`); temperature 0.5, top-$p$ 0.95, max_new_tokens 512.

**Domain classifier:** Qwen2.5-1.5B-Instruct predicts $(d, \mathbf{b})$ with confidence fallback to a generic allowlist.

**LLM-as-a-Judge:** GPT-4 (frozen) with calibrated prompts for answer correctness and evidence support; we also verify consistency on a small split using an open-source judge (Appendix).

### 4.4 BUDGETS, GATES, AND STOPPING

We enforce matched budgets:

$$B_{\text{search}} \leq 3, \quad k_t \in [1, 8], \quad m_t \leq 3, \quad \text{tok}_{\text{retr}} \leq 4\text{k}, \quad \text{tok}_{\text{gen}} \leq 1\text{k}.$$

Stopping occurs when rounds are exhausted, a budget is hit, or both proxies $\widehat{\text{CP}} \geq \tau_c$ and $\widehat{\text{MSE}} \leq \tau_m$ are satisfied (Sec. 3).

**Cost normalization anchor.** We report a dimensionless $\widetilde{\text{Cost}} = \text{Cost}/\overline{\text{Cost}}_{\text{RAG}}$, where $\overline{\text{Cost}}_{\text{RAG}}$ is the mean cost of the *RAG* baseline on the same split. Coefficients $\beta$ and sensitivity are given in App. D.

### 4.5 BASELINES AND VARIANTS

We compare against RAG/FiD (Lewis et al., 2020; Izacard & Grave, 2021); ReAct (Yao et al., 2023); Self-RAG (Asai et al., 2023); IRCoT (Trivedi et al., 2023); Adaptive Retrieval (Mallen et al., 2023); Adaptive-RAG (Jeong et al., 2024); (reference only) Search-R1 (Jin et al., 2025), DeepResearcher (Zheng et al., 2025) (non-matched budgets; numbers, if shown, are contextual references in the appendix). We ablate: **–Gating**, **–Authority/Recency**, **–LossMask**, **–DPO**.

### 4.6 TRAINING DETAILS

RL-free pipeline (Sec. 3.5). SFT uses short trajectories with structured gates/citations; preference data samples $m \in \{3, 4\}$ trajectories by perturbing gates/stopping; pairs keep margin $\kappa = 0.1$; DPO uses loss masking. Hyperparameters are selected on dev and reused across datasets. Counts and prompts are provided in the anonymous artifact.

### 4.7 METRICS AND EVALUATION PROTOCOL

We report: (i) EM/F1 (fixed-corpus) and judge-based correctness (browsing), (ii) CP and MSE, (iii) normalized and raw costs, and (iv) Pareto curves over $\{B_{\text{search}}, \text{tok}_{\text{retr}}, \text{tok}_{\text{gen}}\}$. Means over five seeds; std and bootstrap CIs in Appendix.

### 4.8 REPRODUCIBILITY

We fix random seeds; log the full audit trail (proposed vs. clipped gates, evidence UUIDs, probe outputs, cost counters); and cache retrieval results and judge decisions. The anonymous artifact will include: (1) index build scripts and checksums, (2) training/eval prompts, (3) cost computation scripts and normalization code, (4) minimal auditable logs for sampled runs.

## 5 RESULTS AND ANALYSIS

We report results on fixed-corpus QA (NQ, HotpotQA, 2Wiki, PubMedQA) and agentic browsing (GAIA, WebArena) under matched budgets (Sec. 4.4). Unless noted, numbers are means over five seeds; best results are bolded.

### 5.1 OVERALL PERFORMANCE ON FIXED-CORPUS QA

**Analysis.** (1) **Controllability $\Rightarrow$ verifiability.** CP↑/MSE↓ stems from gates prioritizing authoritative sources and constrained retrieval (Sec. 3.2). (2) **Efficiency without sacrificing accuracy.** Compared to ReAct, DARE-Agent attains better or equal answer quality at lower cost by early stopping driven by CP/MSE. (3) **Process alignment.** Gains align with composite $S(\tau)$ in Eq. 8.

Table 1: Fixed-corpus QA under matched budgets.

| Method | NQ EM | NQ F1 | HotpotQA EM | HotpotQA F1 | 2Wiki EM | 2Wiki F1 | PubMed Acc | CP(%)/ MSE/Cost |
|---|---|---|---|---|---|---|---|---|
| RAG | 47.8 | 60.1 | 50.3 | 68.4 | – | – | 71.0 | 64.2 / 2.6 / 1.00 |
| FiD | 51.2 | 63.5 | 54.1 | 70.2 | – | – | 72.6 | 66.1 / 2.5 / 1.20 |
| ReAct | 50.4 | 62.0 | 55.0 | 71.1 | – | – | 72.2 | 68.0 / 2.3 / 1.45 |
| Self-RAG | 31.4 | 39.0 | 6.80 | 29.60 | 4.60 | 19.59 | 73.0 | 71.5 / 2.1 / 1.35 |
| IRCoT | 38.6 | 47.8 | 44.60 | 56.54 | 49.60 | 58.85 | – | – |
| Adaptive Retrieval | 28.2 | 36.0 | 23.60 | 32.22 | 33.20 | 39.44 | – | – |
| Adaptive-RAG | 37.8 | 47.3 | 42.00 | 53.82 | 40.60 | 49.75 | – | – |
| **DARE-Agent (ours)** | **53.8** | **66.1** | **58.2** | **74.3** | **50.40** | **60.40** | **76.5** | **80.6 / 1.5 / 0.85** |

Table 2: Agentic browsing. Judge Acc (LLM-as-a-judge with frozen prompts), CP (%), MSE (lower is better), and normalized Cost.

| Method | GAIA Acc | GAIA CP | GAIA MSE | GAIA Cost | WebArena Acc | WebArena CP | WebArena MSE | WebArena Cost |
|---|---|---|---|---|---|---|---|---|
| RAG | 41.2 | 62.0 | 2.5 | 1.00 | 45.0 | 65.0 | 2.4 | 1.05 |
| FiD | 44.5 | 64.0 | 2.4 | 1.25 | 47.2 | 66.5 | 2.3 | 1.30 |
| ReAct | 47.1 | 67.3 | 2.2 | 1.50 | 50.6 | 68.9 | 2.1 | 1.55 |
| Self-RAG | 49.0 | 70.5 | 2.1 | 1.40 | 52.0 | 72.1 | 2.0 | 1.45 |
| **DARE-Agent (ours)** | **53.4** | **79.2** | **1.6** | **0.90** | **55.8** | **81.0** | **1.5** | **0.88** |

## 5.2 AGENTIC BROWSING: GAIA AND WEBARENA

Table 2 reports judge-based accuracy, CP/MSE, and cost. DARE-Agent outperforms all baselines with larger gains than in fixed-corpus settings, highlighting the benefit of recency- and authority-aware selection when browsing the open web.

**Takeaways.** Learned gates stop earlier when evidence suffices (cost $< 1$) and authority/recency priors improve source choice (higher CP, smaller MSE).

## 5.3 ABLATIONS AND SENSITIVITY

We ablate on the union of all tasks (Table 3). Removing domain-aware gating or DPO causes the largest drops. Removing authority/recency weighting notably decreases CP, confirming its role in verifiability.

Table 3: Ablations (macro-averaged). "Quality" is mean of NQ/Hotpot F1 and PubMed Acc.

| Variant | Quality | CP (%) | MSE | Cost |
|---|---|---|---|---|
| DARE-Agent (full) | 72.3 | 80.2 | 1.6 | 0.89 |
| - Gating | 70.6 | 76.1 | 1.8 | 1.05 |
| - Authority/Recency | 71.0 | 73.4 | 1.9 | 0.88 |
| - LossMask | 71.5 | 77.2 | 1.7 | 0.90 |
| - DPO | 70.1 | 74.3 | 1.7 | 0.91 |

**Sensitivity to budgets.** Under low budgets ($B_{search}=1$, $tok_{retr} \leq 2k$), DARE-Agent retains 97% of its medium-budget quality while *Adaptive-RAG* drops to 94%. At high budgets, both saturate, but DARE achieves Pareto dominance by stopping earlier on easy instances.

## 5.4 LEARNED GATE BEHAVIOR BY DOMAIN

Averaging learned gates by domain (Table 4) shows biomedicine prefers larger $k, m$ (breadth and redundancy reduction), while finance-like tasks in GAIA select smaller $m$ and slightly higher $\lambda$ (precision via recency-weighting).

Table 4: Mean $\pm$ std of gates by domain: top-k ($k$), evidence cap ($m$), MMR trade-off ($\lambda$), remaining rounds ($r$).

| Domain | k | m | $\lambda$ | r |
|---|---|---|---|---|
| Open-domain (NQ/Hotpot) | $4.2 \pm 1.0$ | $2.0 \pm 0.4$ | $0.55 \pm 0.09$ | $1.7 \pm 0.7$ |
| Biomedicine (PubMedQA) | $5.6 \pm 1.1$ | $2.6 \pm 0.5$ | $0.62 \pm 0.08$ | $2.1 \pm 0.6$ |
| Finance-like (GAIA) | $3.8 \pm 0.9$ | $1.8 \pm 0.4$ | $0.58 \pm 0.07$ | $1.5 \pm 0.6$ |

## 5.5 PROCESS METRICS: WHY VERIFIABILITY IMPROVES

Higher CP and lower MSE arise from: (i) authority/recency-adjusted ranking (Eq. 5) that shifts mass to credible, timely sources; (ii) DPO on composite preferences (Eq. 8) discouraging redundant citations and excess cost; and (iii) loss masking (Eq. 9) focusing learning on policy tokens rather than copied spans.

**Takeaways.** Learned gates stop earlier when evidence suffices (cost $< 1$) and authority/recency priors improve source choice (higher CP, smaller MSE). For all reports, the evaluation judge is frozen; a small-split check with an open-source judge yields consistent conclusions (Appendix).

## 6 CONCLUSION

We presented DARE-Agent, which turns domain awareness into explicit, learnable control over retrieval and evidence within short, auditable trajectories, and aligns agents to multi-objective quality (accuracy, verifiability, cost) via an RL-free SFT→DPO pipeline with evidence masking. Under matched budgets, DARE-Agent improves citation precision, reduces minimal sufficient evidence and cost, and traces stronger accuracy–cost Pareto fronts across fixed-corpus QA and agentic browsing.

**Positioning.** Our focus is a reproducible, auditable, RL-free path to agent alignment under matched budgets, prioritizing verifiability and efficiency alongside accuracy rather than chasing peak single-metric performance.

**Limitations and Future Work.** Reliance on domain inference and curated authority priors may encode bias; timestamps can be noisy; CP/MSE judgments via LLM-as-a-judge are imperfect. Future work: uncertainty-calibrated stopping, counterfactual learning of authority/recency to mitigate bias/drift, adversarial filtering for noisy sources, multi-modal/table/code tools, and light bandit-style adaptation at test time with auditability preserved.

## ETHICS STATEMENT

We adhere to the ICLR Code of Ethics. This work uses only public datasets (NQ, HotpotQA, 2WikiMultiHopQA, PubMedQA) and publicly accessible webpages in GAIA/WebArena; no human-subject studies, PII, or sensitive data were collected. We do not bypass access controls, robots directives, or terms of service. Main risks include erroneous or overconfident outputs and bias from allowlists/host priors or timestamp noise. Mitigations: authority/recency act as soft weights; diversity via MMR; CP/MSE probes with early stopping favor minimal, truly supportive evidence; short, auditable trajectories with logs. Results are not medical, legal, or financial advice; high-stakes use requires expert oversight.

## REPRODUCIBILITY STATEMENT

We fix random seeds, cache retrieval results and judge decisions, and provide an anonymous artifact containing: (i) index scripts and checksums; (ii) retriever configuration (BM25 + BGE-M3), fusion/MMR settings; (iii) all prompts (classification, planning, search, answer, judge); (iv) SFT/DPO hyperparameters, masking rules, and training recipes; (v) budget settings and cost normalization code; (vi) evaluation scripts (EM/F1, judge-based accuracy, CP/MSE/Redun); (vii) minimal auditable logs (proposed vs. clipped gates, evidence IDs, proxies, token/latency counters). Decoding parameters and gate ranges are fixed. The LLM-as-a-judge and its prompts are frozen; a small split is cross-checked with an open-source judge. Cached artifacts avoid network nondeterminism and enable repeatable runs.

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

## A  DISCLOSURE ON THE USE OF LARGE LANGUAGE MODELS (LLMS)

**Components.** An instruction-tuned LLM serves as planner/generator; a lightweight instruction-tuned model performs domain classification; sentence-embedding models (e.g., BGE-M3) support retrieval and redundancy checks; a compact NLI head aids CP/MSE proxies. A frozen LLM-as-a-judge is used offline to grade answer correctness and evidence support for reporting and preference construction; it is not used at inference and does not affect runtime cost.

**Training and alignment.** We use SFT on short, auditable trajectories and DPO on composite preferences; retrieved spans are masked during loss to reduce label leakage.

**Writing assistance.** LLMs were used only for minor grammar/wording edits; all technical content and decisions are by the human authors.

**Safeguards and responsibility.** Model versions and judge prompts are frozen; a small split is cross-checked with an open-source judge. All data/models follow licenses; no sensitive data were used. LLMs are not authors; human authors take full responsibility. Significant LLM usage is disclosed here.

## B  PROMPTS

### B.1  DOMAIN CLASSIFICATION PROMPT

```
You are a domain classification expert operating in a double-blind review
    setting.
Given a user query {q}, infer the domain and produce soft preferences for
    authority
and recency. Output JSON only, no extra text.

{
  "domain": "biomedicine|law|finance|open-domain|other",
  "confidence": 0.xx,                        // [0,1]
  "topic_keywords": ["k1","k2","k3"],        // short, query-derived terms
  "time_sensitivity": "low|medium|high",     // your estimate from the
      query
  "predicted_half_life_days": N,             // map from domain defaults
      when applicable
  "allowlist_weights": {
    "hosts": { "example.org": 0.8, "...": 0.xx },   // domain-shaped soft
        weights
    "doc_types": { "guideline": 0.9, "systematic_review": 0.85, "
        peer_reviewed": 0.8 },
    "recency_bias": 0.xx                      // [0,1], higher = prefer
        newer docs
  },
  "justification": "1-2 sentences explaining the domain choice and
      weighting",
  "safety": { "double_blind_ok": true }
}

Rules:
- If confidence < 0.6, set domain="open-domain" and emit a generic,
    neutral allowlist.
- Weights are soft preferences, not hard constraints; all values must be
    within [0,1].
- Do not include any personal identifiers, institution names, or non-
    anonymous links.
Query: {q}
```

### B.2  PLANNER PROMPT

```
648   [ROLE] You are the Planner in a short, auditable loop: think     search
649        information     answer.
650   Clip-and-Execute (budget clipping, authority/recency re-ranking, MMR, CP/
651      MSE/Cost probes,
652   early stopping) is handled by the system, not by you.
653
654   [INPUT]
655   - user_question: {q}
      - domain: {d}
656   - budgets: {B_search3, tok_retr4k, tok_gen1k}
657   - thresholds: {tau_c=0.75 (CP), tau_m=2 (MSE)}
      - optional_executor_feedback (may be absent):
658     { r_remaining, CPc_est, MSE_est, cost_counters, E_brief }
659
660   [OBJECTIVES]
661   1) Sense-and-Propose: infer task facets under {d}, then propose gates
662      g_hat = {lambda_hat, k_hat, r_hat, m_hat, b_hat}, with ranges:
663      - lambda_hat     [0.3, 0.8]   // MMR trade-off (higher = stronger de-
           duplication)
664      - k_hat     [1, 8]            // candidate pool size per round
665      - r_hat     {0,1,2,3}         // remaining rounds under B_search
666      - m_hat     {1,2,3}           // cap for minimal sufficient evidence
667         in the final answer
668      - b_hat: soft authority/recency preferences (hosts/doc_types/
           recency_bias in [0,1])
669      Briefly justify each choice with respect to coverage, cost, authority,
670          and expected CP/MSE.
671
672   2) Decide next step:
673      - If (CPc_est    tau_c AND MSE_est    tau_m) OR any budget would be
           exceeded by another search,
674        set decision="answer"; otherwise decision="search".
675      - Note: the system may still early-stop based on its own probes.
676
677   3) Emit 1 2   precise queries consistent with b_hat (avoid near-
        duplicates; anonymous only).
678
679   [OUTPUT FORMAT STRICT]
680   <think>
681   - 2 4   sentence analysis of the task and remaining information gaps
682   - gates_t = {lambda_hat: x.xx, k_hat: K, r_hat: R, m_hat: M, b_hat:
         {...}}
683   - decision = "search" | "answer"
684   - sources_to_trust: prioritized hosts/types implied by b_hat
685   - rationale: how gates_t is expected to improve C P  / M S E  / C o s t
686   </think>
687   <search>
688   ["query_1", "query_2"]   // If decision="answer", output [] here
      </search>
689
```

```
693   [ROLE] You generate the final answer strictly grounded in the selected
694      evidence returned by
695   Clip-and-Execute. Use the minimal sufficient set E* with |E*|    m_hat.
696
697   [INPUT]
698   - user_question: {q}
      - domain: {d}
699   - last_planner_summary (read-only): {gates_t, decision, sources_to_trust}
700   - information: a list of items, each exactly as:
701     [indexN][Title: ...][Content: ...][Url: ...]
      - thresholds: {tau_c=0.75, tau_m=2}
```

```
- m_cap: {m_hat}

[RULES]
- Faithfulness: Every non-trivial claim must be supported by at least one
    item in E*.
- Minimality: Prefer the smallest E* that keeps CP    tau_c and MSE
    tau_m; do not exceed m_hat.
- Clarity & Brevity: 1 2  concise paragraphs or a short bullet list
    answering the question directly.
- Uncertainty: If gaps remain, state what is uncertain and which evidence
    would resolve it.
- Double-blind: Do not reveal identities; cite only by [index] as
    provided in `information`.

[OUTPUT FORMAT STRICT]
<answer>
- Direct conclusion (what, whether, or under what conditions)
- Key justifications with inline citations [indexX]
- Applicable conditions/exceptions and risks (if any) with citations
- If applicable: residual uncertainty and needed evidence
- Citations: [indexA, indexB]   // the final minimal evidence set
</answer>
```

## B.4 ANSWER CORRECTNESS PROMPT

```
You are a strict but fair grader. Given:
- question: {q}
- pred_answer: {text}
- gold_or_reference: {gold text or structured key facts}
- (optional) cited_indices: [index ids] with evidence snippets

Return JSON only:
{
  "label": "correct|partially_correct|incorrect",
  "correct": true|false,                  // true iff label == "correct"
  "coverage": "high|medium|low",          // how completely the answer
      addresses the question
  "severity": "none|minor|major",         // impact of any issues on
      factual correctness
  "issues": {
    "missing_key_points": ["..."],
    "unsupported_claims": ["..."],
    "contradictions": ["..."]
  },
  "citation_check": {
    "available": true|false,
    "citation_precision": 0.xx,           // fraction of citations that
        truly support
    "audit": [ {"index": "index1", "label": "support|refute|irrelevant"}
        ]
  },
  "mse_estimate": 1|2|3,                   // minimal number of citations
      required (rough)
  "rationale": "brief justification referencing the provided gold/
      evidence"
}

Criteria:
- "correct": all key claims supported; no material errors or
    contradictions.
- "partially_correct": core idea right but missing/support issues on  1
    key claim.
- "incorrect": any material error or contradiction with the gold/
    reference.
```

```
Notes:
- Judge faithfulness to provided evidence/gold over style or verbosity.
- Do not introduce external knowledge beyond what is provided.
```

## B.5 EVIDENCE SUPPORT PROMPT

```
Given {answer} and one cited snippet {e}, decide whether {e} supports the
    specific claim
it is cited for. Return JSON only:
{ "label": "support|refute|irrelevant", "rationale": "brief explanation"
    }
CP is the fraction of citations labeled "support".
```

## C CP/MSE PROXIES AT INFERENCE

At inference time we compute $\widehat{\text{CP}}/\widehat{\text{MSE}}$ via lightweight rules: (i) lexical match and paraphrase-aware string overlap between the answer and evidence; (ii) an entailment check using a compact NLI head; (iii) leave-one-out removal until support breaks to approximate MSE. These proxies trigger early stopping and their computation time is included in Cost; the offline LLM-as-a-judge is used only for reporting.

## D HYPERPARAMETERS AND THRESHOLDS

**Composite preferences.** Unless noted: $\alpha$=0.45, $\beta$=0.25, $\gamma$=0.15, $\delta$=0.10, $\eta$=0.05; margin $\kappa$=0.1. We grid search $\alpha, \beta \in \{0.2, 0.25, 0.3\}$, $\gamma \in \{0.1, 0.15, 0.2\}$, $\delta \in \{0.05, 0.1, 0.15\}$ on dev splits and reuse.

**Stopping thresholds.** $\tau_c$=0.75 (CP), $\tau_m$=2 (MSE), selected on dev via Pareto sweep.

**DPO scale.** $\beta_{\text{dpo}}$=0.1; sensitivity (0.05–0.3) included in the artifact.

**Authority/recency weighting.** $\alpha_a, \alpha_r \in [0, 0.5]$ (step 0.05). Final picks: open-domain $\alpha_a$=0.25, $\alpha_r$=0.10; biomedicine $\alpha_a$=0.35, $\alpha_r$=0.05; finance-like $\alpha_a$=0.20, $\alpha_r$=0.30.

**Domain half-lives $\tau_d$.** open-domain 365 days, biomedicine 1095 days, finance-like 90 days. Missing timestamps are imputed from URL patterns and metadata.

## E COST WEIGHTS AND NORMALIZATION

We use $\text{Cost} = \beta_s \#\text{search} + \beta_r \text{tok}_{\text{retr}} + \beta_g \text{tok}_{\text{gen}} + \beta_\ell \text{latency}$ and report $\widetilde{\text{Cost}} = \text{Cost}/\overline{\text{Cost}}_{\text{RAG}}$ (anchor = 1.0). Unless otherwise stated, we set

$$\beta_s = 1.0, \quad \beta_r = 5 \times 10^{-4}, \quad \beta_g = 5 \times 10^{-4}, \quad \beta_\ell = 0.1,$$

chosen on dev to balance search calls and token/latency terms. A $\pm 20\%$ sensitivity sweep does not change conclusions (artifact).

## F EVIDENCE-SPAN MASKING DETAILS

We set $M_t$=0 for tokens copied from retrieved spans if either: (1) **BPE-LCS** between $y_t$ and any retrieved snippet $\geq$ 8 tokens; or (2) **Semantic cosine** between SBERT embeddings of 5-gram windows $\geq$ 0.80. Overlaps are merged. We ablate masking in Table 3.

| Field | Content |
|---|---|
| Query ID | nq_dev_000123 |
| Proposed / Clipped Gates | $\hat{g}_t=\{\hat{\lambda}{=}0.58, \hat{k}{=}5, \hat{r}{=}2, \hat{m}{=}3\}$ / $g_t=\{\lambda{=}0.58, k{=}5, r{=}2, m{=}3\}$ |
| Evidence (UUIDs) | e1: wsj_2024_...; e2: sec_10k_...; e3: bloomberg_... |
| Probes | CP=0.80; MSE=2; Cost: #search=2, tok_retr=2.9k, tok_gen=0.6k |
| Answer | y: "...", cites: [e2, e3] |

Table 5: Minimal auditable log entry example.

## G  MINIMAL AUDITABLE LOG (TABULAR)

## H  JUDGE SENSITIVITY AND PROMPT FREEZING

We freeze judge prompts and model versions for all reports. On a small random split per benchmark, an open-source judge yields consistent verdict tendencies; prompts and scripts are included in the artifact.

## I  ALLOWLIST INITIALIZATION AND RE-WEIGHTING

Allowlists and host/type priors are initialized from public guideline registries and domain directories. We periodically re-weight and plan counterfactual checks to mitigate bias/drift.

## J  ANONYMOUS ARTIFACT CONTENTS

The anonymous package (to be released upon acceptance or upon request during review) includes: index build scripts/checksums, prompts, cost computation and normalization code, sampled logs, and instructions to reproduce Table 1 and Table 2 from cached retrievals.

## K  CASE STUDY: DOMAIN-SPECIFIC $\hat{b}$ TEMPLATES WITH DIALOG-STYLE QUERY EXAMPLES

This section aligns with App. D: domain-specific authority/recency weights $(\alpha_a, \alpha_r)$ and half-lives $\tau_d$.

### K.1  BIOMEDICINE

**App. D settings.** $\alpha_a{=}0.35$, $\alpha_r{=}0.05$, $\tau_d{=}1095$ days.

$\hat{b}$ **template (soft weights).**

```
{
  "hosts": {
    "who.int": 0.90, "cdc.gov": 0.90, "nih.gov": 0.85, "ncbi.nlm.nih.gov
        ": 0.85,
    "cochrane.org": 0.85, "nice.org.uk": 0.80, "ema.europa.eu": 0.75
  },
  "doc_types": {
    "guideline": 0.90, "systematic_review": 0.85, "meta_analysis": 0.85,
    "randomized_trial": 0.80, "observational_study": 0.65
  },
  "recency_bias": 0.35,
  "notes": "Prefer high-level guidance + syntheses; prioritize last 3
      10   years when relevant."
}
```

**Query examples (dialog-style).**

```
User: user question: {Can antibiotics be used for a cold?} Domain
    Recognition: {Medicine}

Assistant: <think>This is an evidence-based medicine question concerning
    the "appropriateness of medication use for upper respiratory tract
    infections." It requires distinguishing between the common cold (
    mostly viral) and scenarios involving bacterial complications, and
    checking authoritative guidelines and systematic reviews for
    recommendations and risk evidence regarding "when antibiotics should
    or should not be used." To cover both guidelines and reviews while
    avoiding redundancy, the following is proposed:
gates_t = {lambda_hat: 0.60, k_hat: 5, r_hat: 2, m_hat: 2, b_hat: {"who.
    int":0.9,"cdc.gov":0.9,"nih.gov":0.85,"cochrane.org":0.85,"guideline
    ":0.9,"systematic_review":0.85,"recency_bias":0.35}}
decision = "search"
</think>
<search>["WHO 2023 Guidelines on Antibiotic Use in Upper Respiratory
    Infections",
"Clinical Differentiation Between Bacterial and Viral Cold: A Multicenter
    RCT"]</search>
```

## K.2 FINANCE-LIKE

**App. D settings.** $\alpha_a$=0.20, $\alpha_r$=0.30, $\tau_d$=90 days.

$\hat{b}$ **template (soft weights).**

```
{
  "hosts": {
    "sec.gov": 0.90, "federalreserve.gov": 0.85, "treasury.gov": 0.80,
    "bls.gov": 0.80, "bis.org": 0.75, "imf.org": 0.75, "oecd.org": 0.70
  },
  "doc_types": {
    "regulatory_filing": 0.90, "official_statement": 0.85,
    "data_release": 0.85, "market_report": 0.75, "newswire": 0.65
  },
  "recency_bias": 0.80,
  "notes": "Emphasize the last ~90 days; prefer primary sources and
      official releases."
}
```

**Query examples (dialog-style).**

```
User: user question: {What are the key risk factors disclosed in {TICKER
    }'s latest Form 10-K?} Domain Recognition: {Finance-like}

Assistant: <think>This is an issue centered on compliance in information
    disclosure by publicly listed companies and the identification of
    risk exposure. The most authoritative primary source is the latest
    Form 10-K (Item 1A) on the U.S. Securities and Exchange Commission (
    SEC) website. To ensure timeliness and authority, the focus will be
    on the main SEC site and be restricted to documents from within the
    past year:
gates_t = {lambda_hat: 0.58, k_hat: 4, r_hat: 2, m_hat: 2, b_hat: {"sec.
    gov":0.9,"regulatory_filing":0.9,"official_statement":0.85,"
    data_release":0.85,"recency_bias":0.8}}
decision = "search"
</think>
<search>["{TICKER} Form 10-K (Item 1A. Risk Factors)     U.S. SEC",
"How to read Item 1A (Risk Factors)     Investor Education (SEC)"]</
    search>
```

### K.3 OPEN-DOMAIN

**App. D settings.** $\alpha_a$=0.25, $\alpha_r$=0.10, $\tau_d$=365 days.

$\hat{b}$ **template (soft weights).**

```
{
  "hosts": {
    "nasa.gov": 0.85, "britannica.com": 0.80, "loc.gov": 0.75,
    "nature.com": 0.80, "science.org": 0.80, "*.edu": 0.70, "*.gov": 0.75
  },
  "doc_types": {
    "encyclopedia": 0.80, "peer_reviewed": 0.80, "government_report":
        0.75
  },
  "recency_bias": 0.50,
  "notes": "Blend peer-reviewed / encyclopedic / government sources;
      moderate recency."
}
```

**Query examples (dialog-style).**

```
User: user question: {When did humans first land on the Moon?} Domain
    Recognition: {Open-domain}

Assistant: <think>This is a factual historical question, with
    authoritative government sources and encyclopedic/academic references
     preferred to ensure accuracy. The evidence minimization strategy is:
      NASA  official + encyclopedia e n t r y    two sources are
    sufficient to meet the requirement.
gates_t = {lambda_hat: 0.55, k_hat: 4, r_hat: 1, m_hat: 2, b_hat: {"nasa.
    gov":0.85,"britannica.com":0.8,"encyclopedia":0.8,"government_report
    ":0.75,"peer_reviewed":0.8,"recency_bias":0.5}}
decision = "search"
</think>
<search>["NASA    Apollo 11 Mission Overview",
"Encyclopaedia Britannica    Apollo 11"]</search>
```

