# OpenReview forum: "DARE‑Agent: Domain‑Aware, Resource‑Efficient, Evidence‑grounded Agentic RAG"
_ICLR.cc/2026/Conference — Submitted to ICLR 2026_

### Official Review · Reviewer_zRNA · 2025-10-28

**Soundness:** 3
**Presentation:** 3
**Contribution:** 3
**Rating:** 8
**Confidence:** 3

**Summary:**

The paper introduces DARE-Agent, an agentic RAG framework that makes domain awareness a first-class control signal for retrieval and citing, and explicitly optimizes a multi-objective target: answer accuracy, citation precision (CP), minimal sufficient evidence (MSE), and cost. It implements a short, auditable loop (think → search → information → answer) with a propose-and-clip gating mechanism that sets MMR λ, top-k, number of rounds, evidence cap, and authority/recency weights, which an executor clips to safe ranges. Training is RL-free: SFT for cold start followed by DPO on composite preferences; retrieved spans are loss-masked to avoid label leakage. Evaluated on fixed-corpus QA (NQ, HotpotQA, 2Wiki, PubMedQA) and agentic browsing (GAIA, WebArena) under matched budgets, DARE-Agent attains competitive accuracy while raising CP, lowering MSE, and shifting the accuracy–cost Pareto front; the setup emphasizes reproducibility and auditable logs.

**Strengths:**

1. This paper points out the limitations of current RAG systems: unverifiable answers, uncontrolled cost from excessive tool calls and token
usage, and domain-agnostic retrieval that overlooks authority and recency signals. Meanwhile, answer-accuracy-based evaluation cannot reflect these factors well. To this end, this paper proposes DARE-Agent, which targets a reproducible, RL-free path to agent alignment under matched budgets, with short, auditable trajectories and explicit domain-aware gates.
2. DARE-Agent converts domain recognition into concrete, learnable gates (λ, k, r, m, authority/recency weights) with a deterministic propose-and-clip executor—an interpretable twist on agentic RAG and ReAct-style planning.
3. DARE-Agent adds authority/recency-aware reranking fused with MMR diversity control; instruments process probes (CP/MSE/cost) to guide early stopping—tying verifiability to execution rather than post-hoc scoring.
4. Under matched budgets, DARE-Agent consistently improves CP/MSE and cost while maintaining or improving accuracy on both fixed-corpus QA and web-browsing agents; ablations identify gating and DPO as key contributors.

**Weaknesses:**

The paper is well-written about each part of their framework design, experiment seetings and results, and analysis. I don't see an obvious weakness based on my knowledge. Just some minor issues:

1. Reported CP/MSE and correctness rely partly on LLM-as-a-judge. A deeper cross-check with non-LLM judges/human audits would strengthen claims.

2. Since the proposed framework shows gain on different benchmarks, more explainability about the framework would be appreciated. How does each component works and how do they affect the final answer quality? For example, what's the trade-offs between domain-awareness and retrieval diversity—does higher authority/recency weighting reduce novelty or diversity of evidence?

**Questions:**

See weaknesses

---

### Official Review · Reviewer_vY98 · 2025-11-01

**Soundness:** 2
**Presentation:** 1
**Contribution:** 2
**Rating:** 2
**Confidence:** 3

**Summary:**

This paper introduces DARE-Agent, a framework for building retrieval-augmented generation (RAG) agents that balance accuracy, verifiability, and cost-efficiency across specialized domains. The key innovation is a domain-aware gating mechanism that explicitly controls retrieval parameters (MMR trade-off λ, top-k, search rounds, evidence cap, authority/recency weights) through a propose-and-clip architecture. The system uses supervised fine-tuning followed by Direct Preference Optimization (DPO) on a composite preference score: $S(\tau) = \alpha \cdot \text{Acc} + \beta \cdot \text{CP} - \gamma \cdot \text{MSE} - \delta \cdot \text{Cost} - \eta \cdot \text{Redun}$, where CP (citation precision) and MSE (minimal sufficient evidence) are novel process-level metrics. Training employs evidence-span loss masking to prevent label leakage. Experiments on fixed-corpus QA (NQ, HotpotQA, 2Wiki, PubMedQA) and agentic browsing (GAIA, WebArena) show competitive accuracy with improved CP (+9-12 points), reduced MSE (-35%), and lower cost (-15%) compared to baselines under matched budgets.

**Strengths:**

1. **Novel formulation**: Reframing agentic RAG as multi-objective optimization (accuracy + verifiability + efficiency) is novel and is well motivated.
2. **Strong empirical results on verifiability**: +9-12 points in CP and -35% MSE are substantial improvements that matter for real deployments.

**Weaknesses:**

1. **Critical lack of training data transparency**: The paper never discloses training data scale, collection methodology, domain distribution, or quality filtering criteria. This fundamental gap makes contamination assessment impossible and violates reproducibility standards.
2. **Limited domain generalization**: The system requires pre-curated authority allowlists per domain and falls back to generic settings for unseen domains (confidence < 0.6). No experiments evaluate performance on truly held-out domains. The authors should test zero-shot generalization or explicitly scope claims to "supervised multi-domain specialists" rather than implying general adaptation.
3. **Questionable baseline comparisons**: The paper doesn't clarify which results are reproduced versus cited, and excludes recent strong baselines (Search-R1, DeepResearcher) from matched-budget comparison. All baselines should be carefully reimplemented with clear documentation.
4. **Missing sensitivity ablations**: No analysis of composite preference weights (α, β, γ, δ, η), domain classifier threshold (0.6), gate ranges ([1,8]), or stopping thresholds. Results could be fragile to these choices. Hyperparameter sweeps on development data should be provided.
5. **Poor representation**: The writing of this paper is very unclear. It should be further polished.

**Questions:**

1. How did the authors train the gating mechanism? What is the data like?
2. How does the system perform on domains never seen during training?

---

### Official Review · Reviewer_caQH · 2025-11-01

**Soundness:** 2
**Presentation:** 1
**Contribution:** 2
**Rating:** 2
**Confidence:** 3

**Summary:**

This work is trying to advance the Pareto frontier across accuracy, verifiability, and cost rather than optimizing for peak performance on a single metric. It proposes utilizing a domain-aware gating mechanism, DPO with composite evaluation metrics to address the problem.

**Strengths:**

1. The composite preference design considers the citation precision and cost, which is a reasonable design.
2. This work is trying to advance the Pareto frontier across accuracy, verifiability, and cost rather than optimizing for peak performance on a single metric. This objective makes sense.

**Weaknesses:**

1. The motivation for designing the domain-aware gating mechanism is not clear.
2. How the baselines are selected is not clear. It's hard to verify the experimental validity.

**Questions:**

1. Under what circumstances does the domain-aware gating mechanism play a significant role?
2. Why did you choose these baselines for the comparison?

---

### Official Review · Reviewer_G6Mm · 2025-11-01

**Soundness:** 2
**Presentation:** 1
**Contribution:** 2
**Rating:** 2
**Confidence:** 3

**Summary:**

The paper proposes a new agentic RAG framework **DARE-Agent** feature for being Domain-Aware, Resource-Efficient, and Evidence-Grounded. It involves a inference and training pipieline which includes domain recognization, a propose-and-clip gating mechanism. and a SFT+DPO pipeline. The agent learns to control retrieval breadth, evidence count, and source authority through the dessign. It conducts evaluation on both fixed-corpus QA datasets (NQ, HotpotQA, 2Wiki, PubMedQA) and agentic browsing benchmarks (GAIA, WebArena).

**Strengths:**

Resource-Efficient, and Evidence-Grounded are admirable motivations.

**Weaknesses:**

- The writing is very arbitrart overall, with incomplete sentences, figures of very small fonts, lack of many details about the training and experiments, hard to read main result tables, sections with heavy notations and unnecessary low-level details. This makes it very hard to follow the technical details and the experiments.
- Confusing benchmark choices: the paper evaluates on WebArena and GAIA. Could the authors leave more details about the experiments? They are not for testing RAG systems in general. Additionally, many of the tasks in WebArena is not for QA but operational instead. So it's unclear to me how the experiments are conducted, and whether it was evaluating the full set or just a subset.
- Motivation-wise, even though resource-efficiency and grounded evidence are admirable, it seems very unnecessary to introduce the domain awareness given the status quo of current powerful generalist agents, which also makes the framework unnecessarily complex
- More specifically, regarding the domain, it's now only covering "biomedicine|law|finance|open-domain|other" according to the appendix. This looks arbitrary and limited. Tha paper also lacks study about how accurate is the Qwen2.5-1.5B domain classifier, and how does performance degrade under misclassification?
- Regarding the technical contribution, most components already exist in prior work. The main innovation is the structured (while overly complex) integration of these.

**Questions:**

See the weakness. A quick question is the details about the evaluation on WebArena and GAIA.

---

### Meta-Review · Area_Chair_cbW8 · 2025-12-24

**Summary:**

The paper argues that current LLM research agents struggle in specialized domains due to poor verifiability, high and uncontrolled costs, and weak domain-specific retrieval. It reframes agent quality as a multi-objective trade-off among accuracy, evidence quality, and efficiency, and introduces DARE-Agent, a domain-aware, evidence-grounded RAG framework with auditable control over retrieval. Experiments show that DARE-Agent maintains competitive accuracy while improving citation precision, reducing unnecessary evidence, enhancing authority/recency, and achieving better accuracy–cost trade-offs under fixed budgets.

The strengths include 1) reasonable motivation, formulation and design; 2) good empirical results on verifiability. But the concerns include: 1) the writing needs improvement; 2) Confusing benchmark choices; 3) unnecessary to introduce the domain; 4) the novelty is limited; 5) unclear baseline and experimental validity; 6) missing ablations, data transparency; 7) limited domain generalization; 8) missing deeper cross-check with non-LLM judges/human audits would strengthen claims.

No rebuttal was provided, and the AC agrees with most of the reviewers that this paper is not good enough for ICLR yet.

**Reviewer Concerns:**

The strengths include 1) the writing needs improvement; 2) Confusing benchmark choices; 3) unnecessary to introduce the domain; 4) the novelty is limited; 5) unclear baseline and experimental validity; 6) missing ablations, data transparency; 7) limited domain generalization; 8) missing deeper cross-check with non-LLM judges/human audits would strengthen claims.

No rebuttal was provided.

**Reviewer Scores:**

The scores are 2, 2, 2, 8. No rebuttal was provided, and the AC doesn't think the negative reviewers will change scores.

---

### Decision · Program_Chairs · 2026-01-26

Reject